# Diagnostic Accuracy of [^68^Ga]Ga Labeled Fibroblast-Activation Protein Inhibitors in Detecting Head and Neck Cancer Lesions Using Positron Emission Tomography: A Systematic Review and a Meta-Analysis

**DOI:** 10.3390/ph16121664

**Published:** 2023-11-30

**Authors:** Alessio Rizzo, Alberto Miceli, Manuela Racca, Matteo Bauckneht, Silvia Morbelli, Domenico Albano, Francesco Dondi, Francesco Bertagna, Danilo Galizia, Barbara Muoio, Salvatore Annunziata, Giorgio Treglia

**Affiliations:** 1Department of Nuclear Medicine, Candiolo Cancer Institute, FPO–IRCCS, 10060 Turin, Italy; alessio.rizzo@ircc.it (A.R.); manuela.racca@ircc.it (M.R.); 2Nuclear Medicine Unit, Azienda Ospedaliera SS. Antonio e Biagio e Cesare Arrigo, 15121 Alessandria, Italy; albertomiceli23@gmail.com; 3Division of Nuclear Medicine, IRCCS Ospedale Policlinico San Martino, 16131 Genova, Italy; matteo.bauckneht@unige.it (M.B.); silvia.morbelli@unige.it (S.M.); 4Department of Health Sciences (DISSAL), University of Genova, 16131 Genova, Italy; 5Division of Nuclear Medicine, Università degli Studi di Brescia and ASST Spedali Civili di Brescia, 25123 Brescia, Italy; domenico.albano@unibs.it (D.A.); f.dondi@outlook.com (F.D.); francesco.bertagna@unibs.it (F.B.); 6SC Oncologia Area Nord ASL CN1, 12038 Savigliano, Italy; danilo.galizia@aslcn1.it; 7Division of Medical Oncology, Oncology Institute of Southern Switzerland, Ente Ospedaliero Cantonale, 6501 Bellinzona, Switzerland; barbara.muoio@eoc.ch; 8Unità di Medicina Nucleare, GSTeP Radiopharmacy–TracerGLab, Dipartimento di Diagnostica per Immagini, Radioterapia Oncologica ed Ematologia, Fondazione Policlinico Universitario A. Gemelli, IRCCS, 00168 Rome, Italy; salvatore.annunziata@policlinicogemelli.it; 9Clinic of Nuclear Medicine, Imaging Institute of Southern Switzerland, Ente Ospedaliero Cantonale, 6501 Bellinzona, Switzerland; 10Faculty of Biology and Medicine, University of Lausanne, 1011 Lausanne, Switzerland; 11Faculty of Biomedical Sciences, Università della Svizzera Italiana, 6900 Lugano, Switzerland

**Keywords:** head and neck cancer, FAPI, PET, nuclear medicine, oncology, meta-analysis

## Abstract

Several studies have examined the use of positron emission tomography (PET) using [^68^Ga]Ga-radiolabeled fibroblast-activation protein inhibitors (FAPi) across multiple subtypes of head and neck cancer (HNC). The purpose of the present study was to evaluate the diagnostic accuracy of a newly developed molecular imaging approach in the context of HNC through a comprehensive review and meta-analysis. A thorough literature review was conducted to identify scholarly articles about the diagnostic effectiveness of FAP-targeted PET imaging. The present study incorporates original publications assessing the efficacy of this innovative molecular imaging test in both newly diagnosed and previously treated HNC patients. This systematic review examined eleven investigations, of which nine were deemed suitable for inclusion in the subsequent meta-analysis. The quantitative synthesis yielded a pooled detection rate of 99% for primary HNC lesions. Additionally, on a per patient-based analysis, the pooled sensitivity and specificity for regional lymph node metastases were found to be 90% and 84%, respectively. The analysis revealed a statistical heterogeneity among the studies for the detection rate of primary HNC lesions. The quantitative findings presented in this study indicate a favorable diagnostic performance of FAP-targeted PET imaging in detecting primary HNC tumors. In contrast, discordant results concerning the diagnostic accuracy of lymph node metastases were found. However, further multicentric trials are required to validate the efficacy of FAP-targeted PET in this specific group of patients.

## 1. Introduction

The turn of phrase “head and neck cancers” (HNCs) encompasses an extended group of tumors originating from the oral cavity, which includes the lip, tongue, buccal cavity, palate, salivary glands, rhinopharynx, oropharynx, hypopharynx, larynx, and outer neck, particularly thyroid cancer, as well as cancers of unknown primary (HNCUP), which make their appearance with cervical lymph node metastases [1]. Squamous cell carcinomas (SCCs) originating from the mucosal surface of the oral cavity or the throat account for the majority of head and neck malignancies, whereas sarcomas and adenocarcinomas have a lower incidence; finally, adenoid cystic carcinomas (ACCs) are an uncommon subclass of epithelial HNCs with a perineural development pattern and a combined adenoid and cystic histologic phenotype [1].

With more than 650,000 cases and 330,000 fatalities annually, HNCs are the sixth most frequent malignancy globally, accounting for 1.5% of all cancer deaths worldwide. Furthermore, their incidence rates showed a recent increase, especially in younger patients in the USA and Europe [2]. Two different entities can be recognized by their etiologies: the more frequent carcinogen-associated HNCs, strongly related to tobacco and alcohol, and the virus-associated HNCs [1]; the risk factors for the latter are human papillomavirus (HPV—primarily type 16) and Epstein–Barr virus (EBV). Virus-associated HNCs exhibit distinct biological and clinical characteristics from carcinogen-associated ones [3]. The diagnostic evaluation of HNCs is usually complex due to multiple intricate anatomical structures in the affected regions. These structures are located in close proximity to one another, thereby posing a significant challenge in the assessment of such malignancies. Since the examination of the skull base, which contains multiple foramina and related vascular-nervous structures, can be demanding in clinical practice due to the slight variations in density among the analyzed anatomic sites (which hampers the employment of computed tomography (CT)), contrast-enhanced magnetic resonance imaging (MRI) is currently the best instrumental examination to distinguish healthy tissues from invaded structures [4].

Concerning the staging of such malignancies, the standard imaging techniques currently employed include CT, MRI, and 2-[^18^F]-fluorodeoxyglucose ([^18^F]FDG) positron emission tomography (PET) co-registered with CT or MRI. CT is currently the primary instrumental examination for evaluating neck lesions due to its capacity to assess a wide range of tissue types, its extensive accessibility, and its cost-effectiveness. Nevertheless, MRI is a highly favorable modality for visualizing pathological conditions within the intracranial and intraspinal regions due to its ability to provide excellent tissue characterization and precise demarcation of tumor margins, attributed to the high-contrast resolution, particularly in soft tissues. PET/CT with [^18^F]FDG has demonstrated exceptional sensitivity and specificity in identifying concealed, relapsed, and disseminated HNC lesions in newly diagnosed patients as well as in recurring ones. However, it lacks comprehensive anatomical details, hence necessitating correlation with high-resolution CT or MRI [5].

In recent literature, a significant concept that has been emphasized is that cancer is not solely confined to malignant tumor cells. Rather, it is characterized by a fundamental imbalance of the entire cell environment, known as the tumor microenvironment (TME), which is a complex and dynamic system comprising both cellular and non-cellular components [6]. Cancer-associated fibroblasts (CAFs) are a type of stromal cell that does not express epithelial, endothelial, or leukocyte markers and are notably devoid of oncogene mutations [7]. Moreover, it has been observed that CAFs express alpha-smooth muscle actin (a-SMA) and fibroblast-activating protein (FAP) on their cell membrane. CAFs are a crucial element of the TME and are intricately linked to tumor progression and invasion, to the development of distant metastases, and, more importantly, to therapeutic outcomes [8].

The expression of FAP is typically low in normal adult tissues but noticeably increased in areas undergoing tissue remodeling, such as tumors and inflammation. The results indicate that FAP has emerged as a promising candidate for the molecular imaging of various tumors and non-oncological diseases [9,10]. In this regard, FAP inhibitors (FAPi) have been utilized to develop radiopharmaceuticals that can enhance the in vivo expression of FAP through PET imaging, such as [^68^Ga]Ga-FAPi-02 and [^68^Ga]Ga-FAPi-04 [10]. In this setting, various studies have employed PET imaging techniques with radiolabeled FAPi to identify HNC lesions in diverse clinical scenarios [11]. This systematic review and meta-analysis aims to perform a comprehensive evaluation to ascertain the diagnostic performance of FAP-targeting radiopharmaceuticals in detecting HNC lesions in different clinical settings, excluding thyroid cancers. The secondary objective of this article is to gather empirical data comparing the diagnostic efficacy of FAP-targeted PET with other imaging modalities in patients with HNC.

## 2. Materials and Methods

### 2.1. Protocol

The present systematic review and meta-analysis was performed using a predetermined procedure, adhering to the “Preferred Reporting Items for a Systematic Review and Meta-Analysis” (PRISMA 2020 statement) as a guiding framework for its formulation. The comprehensive PRISMA checklist can be found in the Appendix A. There was no previous registration. In the initial phase, a concise inquiry was formulated: Is it feasible to employ radiolabeled FAPi PET imaging to detect head and neck cancer lesions? Moreover, this systematic review used the population, intervention, comparator, and outcomes (PICO) framework to determine the criteria for selecting studies. The measures included patients who had been diagnosed with HNC cancer (population) and had undergone PET with radiolabeled FAPi tracers, either compared to or without standard-of-care imaging (comparator). The outcomes of interest were the evaluation of FAPi uptake in HNC and the detection rate of FAP-guided PET in patients diagnosed with HNC. The literature search, study selection, and quality rating were carried out separately by two investigators, A.R. and G.T. A consensus conference was convened to address and resolve any discrepancies among the reviewers.

### 2.2. Strategy for Literature Research and Information Sources

Following the establishment of the review question, an extensive literature search was performed utilizing two recognized electronic scholarly databases, namely PubMed/MEDLINE and Cochrane Library; this search aimed to track down papers assessing the diagnostic accuracy of FAP-targeting PET in patients diagnosed with HNC, excluding thyroid cancer. In addition, a search was conducted on the ClinicalTrials.gov database to identify ongoing trials concerning this topic. The date of access for this information was October 18, 2023. The research procedure incorporated the following terms: (A) “PET” OR “positron” AND (B) “FAPI” OR “FAP” OR “fibroblast” AND (C) “head” OR “neck” OR “oropharyn*” OR “hypopharyn*” OR “rhinopharyn*” OR “nasopharynx*” OR “laryn*” OR “throat” OR “voice” OR “sinuses” OR “paranasal” OR “nasal” OR “palate” OR “mouth” OR “salivary” OR “lip” OR “oral” OR “tonsil*” OR “tongue” OR “gum” OR “cheek” OR “epiglottis”. There were no restrictions concerning the language or the year the articles were published. Additionally, a thorough examination of the citations from the included publications was conducted to find additional studies that could strengthen the study’s validity. The most recent update to the literature review was put into effect on 18 October 2023.

### 2.3. Eligibility Criteria

The investigators incorporated clinical research papers yielding insights into the use of FAP-targeted PET in the staging and restaging of HNCs. Articles dealing with malignancies other than HNCs, reviews, letters, remarks, editorials, case reports, brief case series, and original documents from different fields on the topic of interest were not included in the research. To ensure that only pertinent studies were included in the meta-analysis, the studies that did not offer sufficient data for pooling the detection rates of FAP-targeted PET in primary tumor assessment as well as sensitivity and specificity in the detection of regional nodal lesions were excluded from the analysis. Additionally, studies that examined tumor types other than HNCs and those that potentially overlapped patient data from other papers were also excluded.

### 2.4. Selection Method

The titles and abstracts of the acquired publications were assessed based on the predetermined eligibility criteria. The ultimate determination regarding the incorporation of the chosen papers was carried out autonomously for both the systematic review and meta-analysis.

### 2.5. Process of Data Collection and Data Extraction

In order to mitigate potential biases, the researchers independently collected each of the papers included in the analysis and extracted data from the entirety of the manuscripts. The data extraction process involved obtaining relevant information from each study in the systematic review. The present systematic review provided comprehensive information, including the authors’ names, the country of origin, the year of publication, the methodology employed, and funding sources. Additionally, it included specific patient details: sample size, gender, age, clinical setting, and any additional instrumental examinations conducted. Furthermore, the current study outlined the details of the index test, including the administered radiopharmaceuticals, the type of hybrid imaging procedure employed, the patient preparation process, the administered activity, and the time interval between the administration of radiolabeled FAPi and image acquisition.

### 2.6. Quality Assessment (Risk of Bias Assessment)

The QUADAS-2 instrument, utilized for assessing the quality of research on the precision of diagnostic methods, was employed to examine the potential for bias in individual studies and their pertinence to the review question. The authors independently evaluated the quality of the studies included in the systematic review and meta-analysis. The research investigated four domains, namely patient selection, index test, reference standard, and flow and time, for potential bias risk. Additionally, three sectors were reviewed for applicability, including patient selection, index test, and reference standard.

### 2.7. Effects Metrics

The major outcomes of the meta-analysis were the detection rate of FAP-guided PET in HNC primary tumors as well as the sensitivity and specificity in assessing metastatic lymph nodes. The qualitative synthesis (systematic review) analyzed the secondary outcome measures by considering the information provided in the Section 3 of the included papers.

### 2.8. Statistical Analysis

The calculation of the preconceived outcomes was performed using a per patient-based methodology. In accordance with the recommendations put forth by DerSimonian and Laird, the researchers conducted a combined analysis to examine the detection rate of FAP-targeted PET in primary HNC lesions. Additionally, they performed a combined analysis to assess the sensitivity and specificity of FAP-targeted PET in detecting nodal metastases. The data from the included studies were utilized, considering each study’s relative importance, employing a random-effect statistical model. Furthermore, the study included the provision of 95% confidence interval values, which were subsequently visually represented through forest plots. The I-square index, also known as the inconsistency index, was employed to assess the level of statistical heterogeneity within the papers included in the analysis. Statistical heterogeneity was considered significant if the I-square index exceeded 50%. Publication bias was assessed by visually inspecting the symmetry or asymmetry of the funnel plot. The calculations for detection rates were performed using MedCalc^®^ statistical software (version 18.2.1, bvba, Ostend, Belgium). Additionally, the software OpenMeta[Analyst]^®^ (version 3.13), supported by the Agency for Healthcare Research and Quality (AHRQ) in Rockville, MD, USA, was utilized to calculate the pooled values of sensitivities and specificities.

### 2.9. Additional Analyses

Subgroup analyses were performed after identifying statistically significant heterogeneity within the encompassed studies considering study design, patient characteristics, technical factors, and the clinical contexts under investigation.

## 3. Results

### 3.1. Literature Search and Study Selection

The thorough literature search yielded 170 records. According to the information in the Section 2, these 170 publications were scrutinized for eligibility based upon preconceived criteria for inclusion and exclusion, and 159 documents were disqualified (not in the topic of interest, i.e., including patients with tumors other than HNCs, as case reports or reviews). The eleven remaining studies were assessed as suitable for inclusion in the systematic review (qualitative synthesis); nine of them were finally assessed as suitable for meta-analysis (quantitative synthesis) in the following full-text evaluation [12,13,14,15,16,17,18,19,20,21,22]. No further research matching the inclusion criteria emerged after reviewing these articles’ references. Figure 1 summarizes the studies’ selection process.

### 3.2. Study Characteristics

The eleven studies meeting the criteria for inclusion in the systematic review (qualitative analysis), which included 292 HNC patients, are thoroughly analyzed in Table 1, Table 2 and Table 3 [11,12,13,14,15,16,17,18,19,20,21]. The selected studies were published from 2020 to 2023 in China (5/11), Germany (5/11), and Thailand (1/11). Seven included studies accounted for a prospective design [13,16,17,18,19,21,22], while the other four retrospectively analyzed their casuistries [12,14,15,20]. All the included papers reported single-center conduction of the experimentation [12,13,14,15,16,17,18,19,20,21,22]; moreover, ten of the eleven eligible studies disclosed financing resources in the text [12,14,15,16,17,18,19,20,21,22].

According to Table 2, the number of enrolled HNC patients in each study ranged from 8 to 77; their mean/median age ranged from 51.2 to 68.5 years, and the percentage of men varied from 33% to 89%. In four articles, the index test was solely used for staging [13,14,17,21]; in five studies, it was used for both staging and restaging [16,18,19,22]; and in the final two studies, it was used for radiotherapy planning [12,20]. Concerning histologic subtypes, nine publications enlisted different subtypes of HNC [12,13,14,16,17,18,19,20,22], one chose solely ACCs [15], and one focused on nasopharyngeal carcinoma (NPC) [21]. In this context, the HPV status was given in the findings section of two investigations [14,20]. Finally, nine papers compared the outcomes of the index test with [^18^F]FDG PET (co-registered with CT or MRI) [13,14,16,17,18,19,20,21,22], and the remaining two used CT and MRI as comparators [12,15]. Table 2 presents all data related to tumor locations, pathology, and comparative imaging.

The index test characteristics varied significantly between the included studies, as shown in Table 3 of the report. Two studies administered [^68^Ga]Ga-DOTA-FAPi-46 [18,20], one experimentation used both [^68^Ga]Ga-DOTA-FAPi-04 and [^68^Ga]Ga-DOTA-FAPi-74 [15], and two papers did not specify the radiopharmaceutical form of radiolabelled FAPi in the text [12,16]; in the remaining six investigations, the administered radiopharmaceutical was solely [^68^Ga]Ga-DOTA-FAPi-04 [13,14,17,19,21,22]. When evaluated using absolute values, the administered activity ranged from 106.9 to 147 MBq, and when assessed using relative values, it went from 1.85 to 3.5 MBq/Kg. Additionally, there was a 10 to 180 min uptake period between the delivery of radiolabelled FAPi and the PET imaging. Ten investigations employed PET/CT as a hybrid imaging compound [12,13,14,15,17,18,19,20,21,22], whereas one study co-registered PET images with MR [16]. While analyzing PET images, qualitative and semiquantitative analyses were carried out in all the included studies [12,13,14,15,16,17,18,19,20,21,22]. The analyzed semiquantitative variables assessed in the included articles were the target-to-background uptake ratio (TBR), gross tumor volume (GTV), and maximal and mean standardized uptake values (SUVmax and SUVmean) of the examined lesions. The background regions used for TBR measurements differed amongst the included studies: contralateral healthy tissue [14,18,19,21,22], mediastinal blood pool [14], brain [12], oral mucosa [12], muscles [12,17], salivary glands [12], and muscles [12,17].

### 3.3. Risk of Bias and Applicability

The overall assessment of the risk of bias and concerns about the applicability of the included papers according to QUADAS-2 is provided in Figure 2.

### 3.4. Results of Individual Studies (Qualitative Synthesis)

When reported, none of the included studies accounted for adverse effects after administering FAP-targeting radiopharmaceuticals [12,13,17]. Moreover, only one of the included papers assessed the inter-reader agreement of FAP-targeted PET images, stating perfect inter-reader reproducibility (κ values: 0.823, 0.800, and 0.823, *p* < 0.001) [19].

All the reviewed papers assessed variable FAP-targeting radiopharmaceuticals uptake in primary and metastatic HNC lesions; in most reports, it was higher than the uptake of the surrounding healthy tissues [12,13,14,15,16,17,18,19,20,21,22]. Concerning the semiquantitative metrics, average SUV_max_ reported values ranged from 8.7 and 20.8 for primary lesions and varied between 4.3 and 15.4 for metastatic lesions, including local lymph nodes and distant metastases; the high heterogeneity observed among the included studies may be explained by the employment of different FAP-targeting radiopharmaceutical forms as well as the employment of different PET devices. Since every study used different background regions to calculate TBR, its variability has poor value and was not assessed (the TBR regions utilized by each study were reported in the paragraph “study characteristics”).

Based on the results of the included studies, FAP-targeted PET has optimal accuracy in primary tumor detection, allowing for an improved lesion segmentation in primary HNC lesions undergoing radiation therapy, and has a reasonable detection rate of primary tumors in HNCUP patients. In particular, it was superior to [18F]FDG PET when utilized for radiation therapy planning, bringing about a better delineation of primary tumors and differentiation of pathologic areas from surrounding physiologic tissues and preventing potential overtreatment; moreover, FAP-targeted PET had a better performance in assessing the intra-cranial invasion of primary tumors due to its low background uptake in the brain [12,13,14,15,16,17,18,19,20,21,22].

With regard to the lymph node status assessment in HNC patients, the included studies reported conflicting results. In this context, while compared to [^18^F]FDG PET, FAP-directed PET showed an overall superior performance (in terms of sensitivity, specificity, and accuracy) in only one paper [19] and a comparable sensitivity but a superior specificity in two studies [13,22]. On the other hand, in four experimentations, PET could detect an abnormal uptake of FAP-directed radiopharmaceutical in an inferior number of pathologic lymph nodes compared to [^18^F]FDG [14,16,18,21]; of note, the inferiority of FAP-targeted PET was observed only in a per lesion-based analysis, and not all studies confirmed the diagnostic accuracy of both methods through biopsy. Finally, two studies assessed a comparable performance for both examinations [17,20].

Three of the included studies explored the potential role of FAP-targeted PET in detecting distant metastases in HNC patients. In this context, two of them observed only a slight superiority of FAP-directed radiopharmaceuticals over [^18^F]FDG in revealing the presence of distant metastases, especially in the bone [18,22]; conversely, one study reported concordance between FAP-targeted and [^18^F]FDG PET examinations on all metastatic sites [17]. Since a few of the included studies involved a distant metastasis analysis in their design (each with a slight percentage of metastatic patients), a meta-analysis of the detection rate in this setting could not be accomplished.

Concerning the tie-up between semiquantitative metrics at FAP-targeted PET and pathological characteristics of HNCs, four studies observed an overall high grade of FAP staining with an immunohistochemistry analysis [13,14,21,22]. A single study found a significant relationship between FAP staining in the immunohistochemical analysis and radiolabeled FAPi uptake (measured as SUVmax) [22]. Conversely, another study found no statistically significant correlation between FAP staining grade and PET measurements (measured as SUVmax). [21]. Furthermore, three studies collected the analysis of HPV and EBV status in HNCs patients; nevertheless, none compared PET semiquantitative values with this dichotomic variable [14,17,20].

Five of the included studies reported how often and in which modalities PET/CT with radiolabeled FAPi could affect the management of HNC patients through a stage modification. In particular, Gu et al. reported finding a previously unknown primitive lesion in 7/13 HNCUP patients [17]; Promteangtrong et al. assessed an upstaging in 2/40 patients and downstaging in 1/40 patients [18]; Jiang et al. observed an upstaging in 11/77 patients and a downstaging in 8/33 patients [22]; Zheng et al. revealed that FAP-targeted PET upstaged 13/47 primary tumors, downstaged 8/47 primitive lesions, and underestimated the lymph node stage in 21/47 patients [21]; and Qin et al. evidenced radiolabeled FAP-guided upstaging and downstaging in 3/15 patients, respectively [16].

Finally, one of the collected studies focused on the potential role of FAP-targeted PET in 12 ACC patients using CT and MR as comparators, reporting a more precise primitive tumor segmentation than CT [15]. In this study, radiolabeled FAP-guided PET was more accurate than conventional imaging, detecting more distant metastases in the lungs, pancreas, and peritoneum, upstaging 5/12 enrolled patients. Moreover, the authors observed variable FAP staining with an immunohistochemistry analysis.

A comprehensive overview of the main results of each included paper is synthesized in Table 4.

### 3.5. Meta-Analysis (Quantitative Synthesis)

As mentioned in the Section 2, the meta-analysis was split into two sub-analyses that looked at the detection rate (DR) of FAP-targeted PET/CT in primary HNCs (per patient-based analysis) as well as the sensitivity and specificity in determining local lymph node involvement on a per patient-based analysis. The assessment of the pooled DR for distant metastases was not accomplished due to a lack of data. As stated in the summary of the study selection process (Figure 1), one study exploring the diagnostic performance of radiolabeled FAPi in ACC was excluded from the quantitative analysis since ACC has different pathogenesis, metabolic behavior, and management than other HNCs [15].

#### 3.5.1. Detection Rate of Primary Tumor

For the pooled analysis of the primary tumor DR on FAP-targeted PET imaging, nine studies involving 241 HNC patients were selected [13,14,16,17,18,19,20,21,22]. Overall, the DR of PET with FAPi, co-registered with CT or MRI, for detecting primary HNCs ranged from 38.89% to 100% (Table 5).

The combined DR of primary HNC was 97.8 (95% confidence interval (95%CI): 95.09–99.22) (Figure 3). A high statistical heterogeneity among the included studies was found as I^2^ was 82.67%; finally, the funnel plot for publication bias assessment (Figure 3) showed only one outlier [17], supporting the absence of significant publication biases.

Based on the reported statistical heterogeneity, a subgroup analysis omitting the only study that enrolled patients with unknown primitive was performed [17]. The subgroup analysis showed a pooled DR of 100% (95% CI: 99.21–100) without significant statistical heterogeneity among the included studies, 10 as I^2^ was inferior to 50%.

#### 3.5.2. Sensitivity and Specificity in Lymph Node Metastasis

Six papers that reported on the diagnostic accuracy of FAP-targeted PET in assessing lymph node status in 82 HNC patients were included in this sub-analysis. [13,14,16,17,19,22].

Based on a per-patient analysis, the pooled sensitivity and specificity of PET with radiolabeled FAPi in the local lymph node metastasis assessment were 0.90 (95% CI: 0.81–0.95) and 0.84 (95% CI: 0.61–0.95). The relative summary receiver operating characteristics (SROC) and forest plots are shown in Figure 4 and Figure 5, respectively.

Figure 6 and Figure 7 show the combined negative and positive likelihood ratios as well as the diagnostic odds ratio, which were, respectively, 0.01 (95% CI: 0.04–0.23), 3.84 (95% CI: 1.62–9.06), and 55.34 (95% CI: 12.99–235.81). Since the inconsistency index for the studies in this sub-analysis was continuously below 50%, there was no substantial statistical heterogeneity among them.

## 4. Discussion

The overexpression of FAP on the cell membrane of stromal cells within the TME poses a promising opportunity for molecular imaging and maybe radioligand therapy [10]. In the past few years, there has been a gradual increase in clinical studies dealing with the utilization of FAP-targeted PET imaging in different oncology settings. This emerging body of research provides valuable insights into the potential employments of this novel imaging technique. Moreover, recent investigations have indicated that FAP-targeted PET has demonstrated outstanding results in identifying various malignancies, including neoplasms typically associated with low or negligible levels of [^18^F]FDG uptake [23,24,25]. These advantages in FAP-targeted PET imaging are partially brought about by the relatively low background activity levels of muscle and blood pool, leading to a higher TBR and, thus, superior image quality compared to [^18^F]FDG PET [26].

Within the course of the previous three years, several clinical studies have endeavored to evaluate the diagnostic efficacy of PET imaging administering radiopharmaceuticals targeting FAP in order to assess its diagnostic accuracy in different clinical scenarios and to find out its potential indications in patients with HNCs [12,13,14,15,16,17,18,19,20,21,22]. These investigations have encompassed newly diagnosed patients and those previously undergoing surgical procedures or chemotherapy treatments. The purpose of this meta-analysis was to combine the existing data to enhance their statistical power and obtain a more reliable estimate of the performance of FAP-targeted PET compared to individual studies.

Regarding the diagnostic performance of PET imaging using radiopharmaceuticals targeting FAP in the detection of primary HNC lesions, remarkable precision was shown in both the initial assessment and in a restaging setting. Most studies included in the meta-analysis relied on a comparison between FAP-targeted PET imaging, co-registered with CT or MRI, and [^18^F]FDG PET. These studies consistently showed an optimum primary tumor detection rate in HNC patients [12,13,14,15,16,17,18,19,20,21,22]. In one study, Syed et al. observed a superior performance of FAP-guided PET over contrast-enhanced CT in target volume delineation for radiation therapy planning, proposing an automatically generated target volume segmentation based on different experimental tumor-to-healthy tissue FAPi-SUV ratios; this statement underlies the achievement of high-contrast images obtained through the administration of FAP-targeting radiopharmaceuticals due to specific tracer uptake in pathologic tissues and low background noise, potentially allowing for radiation oncologists to accomplish more effective treatments while reducing the absorbed dose of surrounding healthy tissues [12]. In this setting, more prospective randomized trials with medium-term follow-up are warranted to assess if FAP-guided radiation therapy is a valuable instrument to delineate target volumes in radiation therapy planning, reaching a reduction in radiation-induced side effects and longer progression-free survivals than conventional imaging-based treatments. Furthermore, in one of the included papers, FAP-guided PET could identify more primitive lesions than [^18^F]FDG PET in patients diagnosed with HNCUP [17]. If confirmed by future research, this finding can reduce the employment of invasive diagnostic procedures (such as tonsillectomy) to identify primary tumors in these patients.

Conversely from what was observed for primary tumor assessment, FAP-guided PET showed discordant diagnostic performances compared to [^18^F]FDG PET imaging in detecting lymph node metastases and, despite our meta-analysis, reported a pooled sensitivity and specificity of 90% and 84% on a per patient-based analysis, in four studies FAP-targeted PET could detect fewer pathologic lymph nodes than a [^18^F]FDG scan [14,16,18,21]. Nevertheless, it is worth noting that, out of the four included studies reporting the inferiority of FAP-guided PET in detecting lymph node metastases compared to [^18^F]FDG PET, only one provided histopathological confirmation of the lesions [14]. Based on these data and the notable specificity in a per-lesion analysis reported by two other studies included [13,22], it is not feasible to definitively rule out the possibility that the metastases identified only by [^18^F]FDG PET (or a subset thereof) were false positive findings. Consequently, there is a potential for the FAP-guided PET performance to be underestimated. Additional prospective studies are required in this particular context to comprehensively evaluate FAP-targeted PET’s diagnostic performance in assessing lymph node metastases’ status.

Although a histopathologic analysis is generally considered the most reliable method for evaluating distant metastases, non-invasive imaging has emerged as a significant advancement in staging patients with cancer, including HNCs. Overall, the investigations included in this systematic review demonstrated that FAP-targeted PET imaging exhibited a slightly superior performance compared to [^18^F]FDG PET in identifying distant metastases in common and unusual locations, such as the bone and the lungs [22]. As observed in the recent literature, FAP expression in the TME is a factor promoting cell migration and subsequent distant metastasis onset; once distant metastases are generated, its expression decreases significantly with a consequent reduction in the radiopharmaceutical uptake. This observation might explain the heterogeneity in terms of FAP-targeting radiopharmaceutical uptake in distant metastases since the PET examination might reveal different lesions in different stages of their evolution [27,28]. Unfortunately, a meta-analysis of the detection rate in this setting was not feasible since only a few included studies added the detection rate assessment of distant metastases.

A novel diagnostic examination has the potential to affect a patient’s management solely if its utilization results in a reclassification, either upward or downward, in comparison to traditional imaging, subsequently impacting the course and the choice of therapy. As observed in the included papers, FAP-targeted PET could upstage and downstage a significant percentage of patients in different investigations, especially concerning the lymph node status assessment [16,17,18,21]. In this setting, more studies involving a histopathological analysis of neck lymph nodes are needed. Furthermore, the authors warrant prospective randomized trials to assess if therapies based on FAP-targeted PET could be more cost-effective than conventional imaging-guided treatments, especially considering the high specificity of this novel instrumental examination.

Of the included studies, only one explored the potential role of PET imaging with FAP-targeting radiopharmaceuticals in ACC, reporting a superior performance compared to contrast-enhanced CT in detecting primitive lesions as well as local and distant metastases [15]. Concerning this particular malignancy, recent molecular imaging and immunohistochemistry studies focused on its expression of prostate-specific membrane antigen (PSMA, also known as carboxypeptidase type II), reporting a high expression of this transmembrane protein on the ACC tumor cell membrane, an optimal diagnostic accuracy of PSMA-targeted PET imaging, and the potential for PSMA to be an effective target for radioligand therapy [29,30]. Since the available literature does not provide any comparison between these two molecular imaging techniques and considering the recent introduction of bispecific tracers (targeting both FAP and PSMA), prospective studies involving both diagnostic methods are needed to clearly assess which instrumental examination is more reliable in this kind of malignancy [31]. Regarding molecular imaging in ACC diagnostics, it is noteworthy to acknowledge the existence of an additional tracer, namely [^11^C]methionine. This particular tracer enables the examination of amino acid metabolism in various benign and malignant conditions, including ACC, showing better performances in well-differentiated histologies [32,33]. Although this radiopharmaceutical has demonstrated promising performances in this particular context, its utilization is constrained by the unfeasibility of large-scale manufacturing, primarily due to the inherent physical features of [^11^C]carbon and its relatively short decay period of approximately 20 min. As for the previously mentioned radiopharmaceuticals, none of the studies included in this systematic review and meta-analysis compared the diagnostic performance of FAP-targeted PET to amino acid-based imaging.

Although in the last decade immune checkpoint inhibitors (ICIs) have significantly improved the treatment of different cancer types, including HNCs, there is currently a lack of effective longitudinal biomarkers for predicting patient response and prognosis when treated with ICIs either as a standalone or in combination with chemotherapy [34]. In this particular context, several investigations have shown that the elevated expression of FAP inside the TME could potentially serve as an indicator of worse prognosis in patients diagnosed with different types of cancers treated with ICIs [35]. However, none of the papers included in the current analysis investigated the potential of FAP-targeting PET in predicting treatment outcomes across various treatment regimens. Additionally, none of the studies delved into the potential significance of this innovative imaging technique in evaluating the response to therapy. Given the importance of this subject matter and the increasing volume of research endeavors aimed at identifying prognostic indicators capable of assessing a patient’s potential response to these therapies, further clinical trials should be conducted to investigate the utilization of FAP-targeted PET in this context.

This systematic review and meta-analysis acknowledges certain limitations. To begin with, it is worth noting that a significant proportion, approximately 40%, of the included papers exhibit a selection bias stemming from a limited sample size. This particular characteristic can potentially impact the validity and reliability of the observed findings.

Furthermore, it should be noted that a subset of the studies did not include histological confirmation of FAP-targeted and [^18^F]FDG PET findings, inducing a lack of consensus concerning the evaluation of lymph node status and has resulted in conflicting conclusions.

Heterogeneity among the papers incorporated in a meta-analysis could potentially introduce bias. The present study revealed notable heterogeneity among the included studies in evaluating the detection rate of primary HNCs, as indicated by an inconsistency index exceeding 50%. A subsequent subgroup analysis was conducted, excluding the paper that enrolled patients diagnosed with HNCUP [17]. The findings of this second analysis did not demonstrate any significant statistical heterogeneity.

## 5. Conclusions

The systematic review and meta-analysis presented has furnished both qualitative and quantitative data, which underscore the potential of FAP-targeted PET imaging in identifying primary tumors and distant metastases in patients with HNC. However, further research is required to validate PET imaging with FAP-targeting radiopharmaceuticals to more precisely determine the significance in evaluating lymph node metastases, to better define which TBR regions are more effective to improve a semiquantitative image evaluation, and to explore its potential as a longitudinal biomarker during treatment in order to develop more consistent clinical recommendations.

## Figures and Tables

**Figure 1 pharmaceuticals-16-01664-f001:**
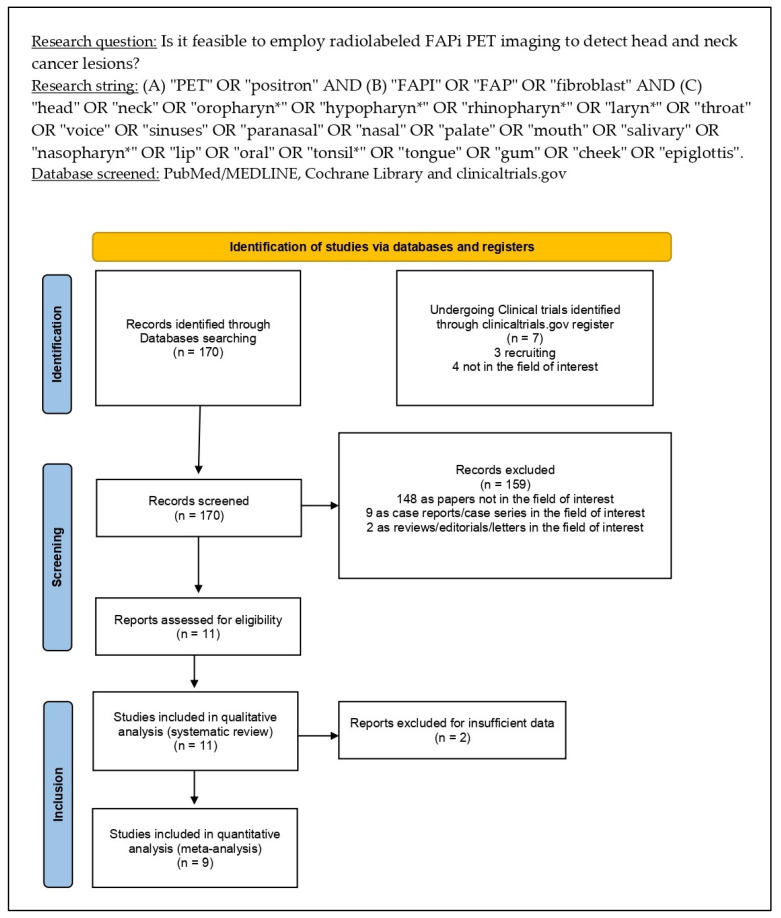
Results of the literature search.

**Figure 2 pharmaceuticals-16-01664-f002:**
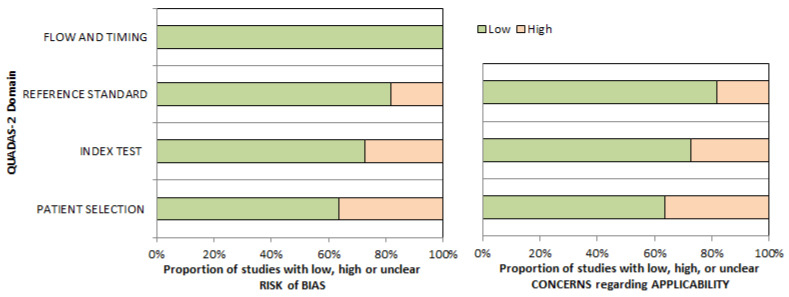
A concise overview of the quality evaluation conducted using the QUADAS-2 tool. The authors categorized the studies included in the systematic review based on their level of bias or applicability issues for specific topics stated on the ordinate axis. Conversely, the abscissa axis represents the proportion of studies. According to the graph, almost 40% of the studies analyzed exhibit a significant risk of bias in the domain of “patient selection.” Conversely, fields such as “reference standard,” “index test,” and “flow and timing” demonstrate a lower risk of bias.

**Figure 3 pharmaceuticals-16-01664-f003:**
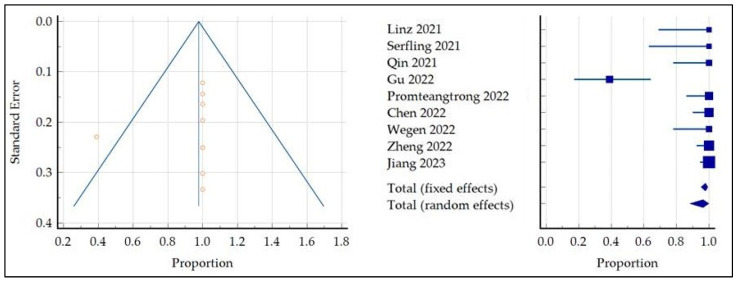
Funnel plot and meta-analysis concerning the DR of FAP-targeted PET in primary HNC lesions [13,14,16,17,18,19,20,21,22].

**Figure 4 pharmaceuticals-16-01664-f004:**
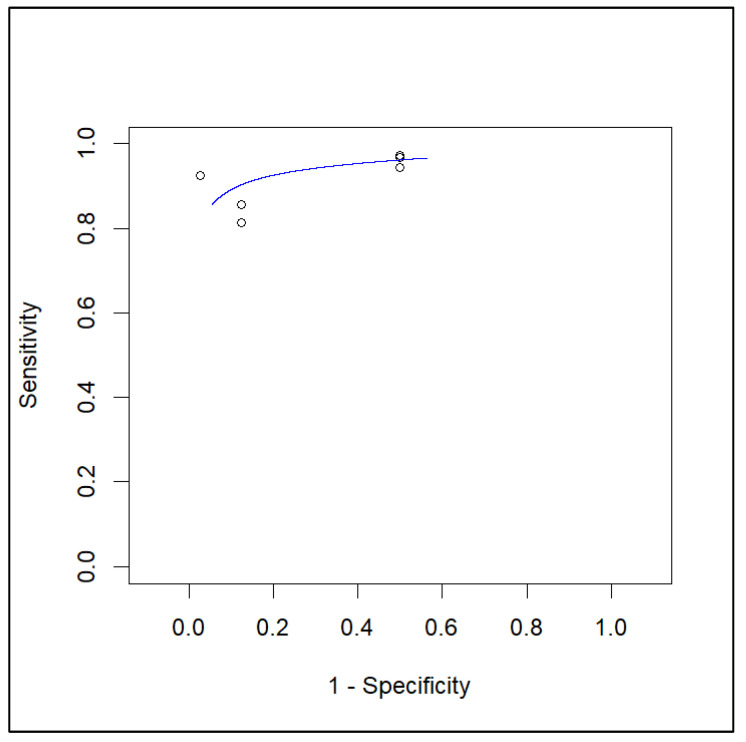
SROC curve of index test’s diagnostic accuracy in lymph node metastasis.

**Figure 5 pharmaceuticals-16-01664-f005:**
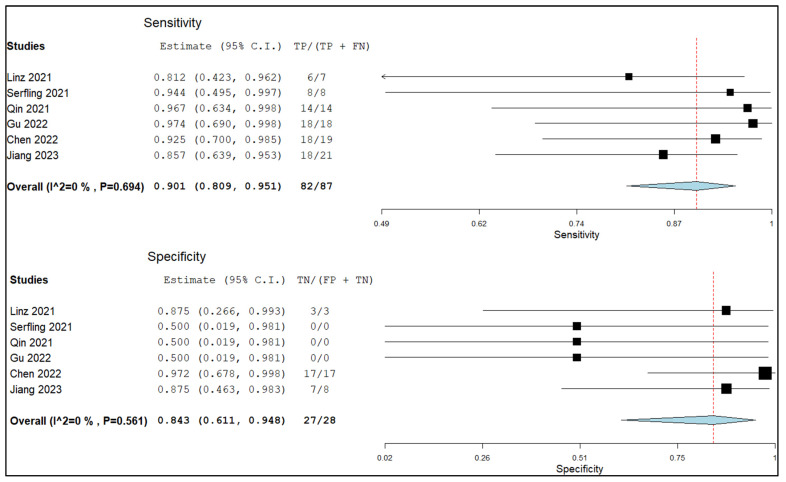
Sensitivity and specificity of the index test in assessing lymph node metastasis and relative forest plots [13,14,16,17,19,22]. Legend: 95% C.I.: 95% confidence interval; TP: true positive; TN: true negative; FP: false positive; FN: false negative.

**Figure 6 pharmaceuticals-16-01664-f006:**
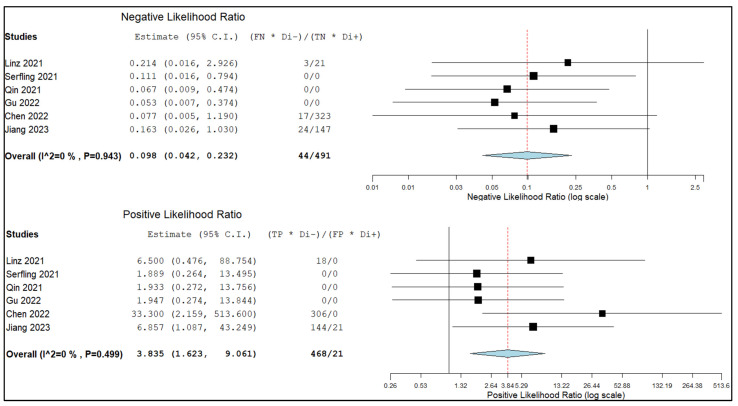
Negative and positive likelihood ratio of the index test in the assessment of lymph node metastasis and relative forest plots [13,14,16,17,19,22]. Legend: 95% C.I.: 95% confidence interval; TP: true positive; TN: true negative; FP: false positive; FN: false negative.

**Figure 7 pharmaceuticals-16-01664-f007:**
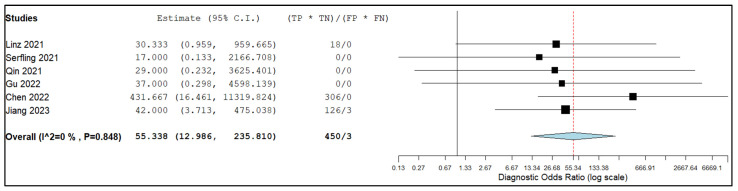
Diagnostic Odds ratio of the index test in assessing lymph node metastasis and relative forest plots [13,14,16,17,19,22]. Legend: 95% CI.: 95% confidence interval; TP: true positive; TN: true negative; FP: false positive; false negative.

**Table 1 pharmaceuticals-16-01664-t001:** General study information.

First Authors [Ref.]	Year	Country	Study Design/Number of Involved Centers	Funding Sources
Syed [12]	2020	Germany	Retrospective/Monocentric	Open Access funded by Projekt DEAL. No funding declared concerning the development of the study.
Linz [13]	2021	Germany	Prospective/monocentric	None declared
Serfling [14]	2021	Germany	Retrospective/Monocentric	Open Access funded by Projekt DEAL. No funding was declared concerning the development of the study.
Röhrich [15]	2021	Germany	Retrospective/Monocentric	Funded by the Federal Ministry of Education and Research (grant number 13N 13341).
Qin [16]	2021	China	Prospective/Monocentric	Funded by the Key Project of the National Natural Science Foundation of China (No. 81630049, 82030052) and National Key R&D Program of China (No. 2019 YFC 1316204).
Gu [17]	2022	China	Prospective/Monocentric	Funded by the National Natural Science Foundation of China (No. 81771861, 81971648, and 81901778) and Shanghai Anticancer Association Program (No. HYXH2021004).
Promteangtrong [18]	2022	Thailand	Prospective/Monocentric	NSofie iTheranostics Inc. provided the 68Ga-FAPI-46 precursor.
Chen [19]	2022	China	Prospective/Monocentric	National Natural Science Foundation of China (No. 81971651, No. 82171928), Natural Science Foundation of Fujian Province (No. 2019J01454, No. 2020J05249), Fujian Provincial Health Technology Project (No. 2020GGA045, No. 2020QNA054) and Startup Fund for Scientific Research of Fujian Medical University (No. 2017XQ1099)
Wegen [20]	2022	Germany	Retrospective/Monocentric	Open Access funded by Projekt DEAL. No funding was declared concerning the development of the study.
Zheng [21]	2022	China	Prospective/Monocentric	National Natural Science Foundation of China (No. 81971651)
Jiang [22]	2023	China	Prospective/Monocentric	National Natural Science Foundation of China (No. 82171986); Improvement Project for Theranostic Ability on Miscellaneous Disease (ZLYNXM202007); fundamental research funds for the central universities, Wuhan University (2042021kf0160); research fund from medical Sci-Tech innovation platform of Zhongnan Hospital, Wuhan University (PTXM2021021).

**Table 2 pharmaceuticals-16-01664-t002:** Patient key characteristics and clinical settings.

First Authors [Ref.]	Sample Size (No. of Patients)	Mean/Median Age (Years)	Gender (Male %)	No. of Patients and Clinical Setting	Location of Primary Tumor (No. of Patients)	HNC Subtype (No. of Patients)	Comparative Imaging
Syed [12]	14	Median: 68.5	86%	14 RT planning	n.a.	12 SCC 1 Mucoepidermoid carcinoma 1 Undifferentiated	CT; MRI
Linz [13]	10	Mean: 62	80%	10 Staging	5 Mouth floor 2 Tongue 2 Mandible alveolar process 1 Maxillary mucosa	10 SCC	[^18^F]FDG PET/CT; MRI
Serfling [14]	8	Mean: 62	75%	8 Staging	8 Waldeyer’s tonsillar ring	8 SCC (6 HPV+)	[^18^F]FDG PET/CT
Röhrich [15]	12	Mean: 57.8	33%	7 Staging 5 Restaging	12 Salivary glands	12 ACC	CT; MRI
Qin [16]	15	Mean: 51.2	53%	14 Staging 1 Restaging	15 Nasopharynx	3 K SCC 2 nK SCC 8 nK undifferentiaded carcinoma 2 Unknown	[^18^F]FDG PET/MRI
Gu [17]	18	Median: 55	89%	18 Staging	18 HNCUP	16 SCC 2 ADC	[^18^F]FDG PET/CT
Promteangtrong * [18]	40	Mean: 57	68%	12 Staging 28 Restaging	17 Nasopharynx 10 Tongue 5 Pyriform fossa 1 Lip 1 External ear canal 1 Nasal cavity 1 Oropharynx 1 Retromolar trigone 1 Oral mucosa 1 Glottis 1 Mouth floor	40 SCC	[^18^F]FDG PET/CT
Chen ** [19]	36	Mean: 62	81%	17 Staging 8 Restaging	15 Tongue 6 Floor of the mouth 5 Buccal mucosa 5 Gingiva 3 Root of the tongue 3 Palate	36 SCC	[^18^F]FDG PET/CT
Wegen [20]	15	Median: 66	80%	15 RT planning	3 Nasopharynx 8 Oropharynx 1 Hypopharynx 3 Larynx	14 SCC (5 HPV+) 1 ADC	[^18^F]FDG PET/CT
Zheng [21]	47	Mean: 52	68%	47 Staging	47 Nasopharynx	47 NPC (30 EBV+)	[^18^F]FDG PET/CT; MRI
Jiang [22]	77	Median: 58	79%	67 Staging 10 Restaging	18 Nasopharynx 19 Oral cavity 14 Oropharynx 16 Larynx 7 Hypopharynx 3 Nasal cavities and paranasal sinuses	77 SCC	[^18^F]FDG PET/CT

* Out of the 40 patients enrolled, 18 patients were included in the analysis for diagnostic accuracy. ** One patient had two primitive lesions in the tongue and gingiva. Legend: ACC: adenoid-cystic carcinoma; ADC: adenocarcinoma; CT: computed tomography; EBV: Epstein–Barr virus; FDG: fluorodeoxyglucose; HNCUP: head and neck cancer of unknown primitive; HPV: human papillomavirus; K: keratizing; MR: magnetic resonance; n.a.: not available; nK: non-keratizing; PET: positron emission tomography; RT: radiotherapy; SCC: squamous cell carcinoma.

**Table 3 pharmaceuticals-16-01664-t003:** Index test key characteristics.

First Authors [Ref.]	Tracer	Hybrid Imaging	Tomograph	Administered Activity	Uptake Time(Minutes)	Image Analysis
Syed [12]	[^68^Ga]Ga-FAPi (pharmaceutical form not specified)	PET/CT	Biograph mCT Flow (Siemens ^®^)	n.a.	30	Qualitative, semiquantitative (SUV_max_, SUV_mean_, GTV, TBR)
Linz [13]	[^68^Ga]Ga-DOTA-FAPi-04	PET/CT	Biograph mCT 64 (Siemens ^®^)	Mean: 119 MBq	n.a.	Qualitative, semiquantitative (SUV_max_, SUV_peak_)
Serfling [14]	[^68^Ga]Ga-DOTA-FAPi-04	PET/CT	Biograph mCT 64 (Siemens ^®^)	Mean: 145 MBq	60	Qualitative, semiquantitative (SUV_max_)
Röhrich [15]	[^68^Ga]Ga-DOTA-FAPi-04; [^68^Ga]Ga-DOTA-FAPi-74	PET/CT	Biograph mCT Flow (Siemens ^®^)	n.a.	10; 60; 180	Qualitative, semiquantitative (SUV_max_, SUV_mean,_ GTV)
Qin [16]	[^68^Ga]Ga-FAPi (pharmaceutical form not specified)	PET/MRI	SIGNA(GE Healthcare ^®^)	1.85–3.7 MBq/kg	30–60	Qualitative, semiquantitative (SUV_max_)
Gu [17]	[^68^Ga]Ga-DOTA-FAPi-04	PET/CT	Biograph mCT flow scanner (Siemens ^®^)	Mean: 143.71 MBq	60	Qualitative, semiquantitative (SUV_max_, SUV_mean_, TBR)
Promteangtrong [18]	[^68^Ga]Ga-FAPi-46	PET/CT	Biograph Vision (Siemens ^®^)	2.0 MBq/kg	60	Qualitative, semiquantitative (SUV_max,_ FTV, TLA, TBR)
Chen [19]	[^68^Ga]Ga-DOTA-FAPi-04	PET/CT	Biograph mCT 64 (Siemens ^®^)	1.85–2.2 MBq/Kg	60	Qualitrative, semiquantitative (SUV_max_, TBR)
Wegen [20]	[^68^Ga]Ga-DOTA-FAPi-46	PET/CT	Biograph mCT Flow–Edge 128 (Siemens ^®^)	Mean: 147 MBq	Following tracer administration	Qualitative, semiquantitative (SUV_max_, SUV_mean_, GTV, TBR)
Zheng [21]	[^68^Ga]Ga-DOTA-FAPi-04	PET/CT	Biograph mCT 64 (Siemens ^®^)	Mean: 106.9 MBq	44	Qualitative, semiquantitative (SUV_max_, TBR)
Jiang [22]	[^68^Ga]Ga-DOTA-FAPi-04	PET/CT	n.a.	n.a.	n.a.	Qualitative, semiquantitative (SUV_max_, TBR)

Legend: CT: computed tomography, DOTA: 1,4,7,10-tetracetic-1,4,7,10-tetraazaciclododecan acid; FAPi: fibroblast activation protein inhibitor, FTV: FAP expression tumor volume; GTV: gross tumor volume; MR: magnetic resonance; n.a.: not available; PET: positron emission tomography; TBR: target-to-background ratio; TLA: total lesion activity.

**Table 4 pharmaceuticals-16-01664-t004:** Outcomes of the included studies.

First Authors [Ref.]	Aim of the Study	Primary Lesion SUV_max_	Metastatic Lesions SUV_max_	Immunohistochemistry for FAP in Tumor Struma	Outcome
Syed [12]	Investigate the use of FAPi PET/CT to detect and delineate HNCs for RT planning.	Mean: 14.6 ± 4.4	Lymph nodes: 9.4 ± 5.7 Bone: 7.5 ± 1.8	n.a.	Tumor segmentation based on RT planning on FAPi PET resulted in larger treatment volumes, including FAPi-avid regions not covered by CT.
Linz [13]	Assess diagnostic accuracy of FAPi PET.	Mean: 20.8 ± 6.4	Lymph nodes: 10.7 ± 6.9	All primary tumors and lymph node metastases showed positive FAP immunostaining.	FAPi PET has superior specificity over FDG PET and might prevent potential overtreatment.
Serfling [14]	Investigate the use of FAPi PET/CT to detect and delineate ACC for RT planning.	Mean: 15.9 ± 6.3	n.a.	Most of the primary tumors and lymph node metastases showed positive FAP immunostaining.	The differentiation between the primary tumor and surrounding physiologic tissues is improved by FAPi PET when compared to FDG PET.
Röhrich [15]	Assess diagnostic accuracy of FAPi PET.	Mean: 12.8 ± 2.1	Mean of all sites: 4.3 ± 1.2	ACC-stroma was variably positive for FAP immunostaining.	FAPi PET allowed a more accurate tumor segmentation than CT.Moreover, FAPi PET revealed additional lesions in 5/12 ACC patients.
Qin [16]	Assess diagnostic accuracy of FAPi PET.	Mean: 13.9 ± 5.1	Lymph nodes: 8.8 ± 3.8	n.a.	FAPi PET is superior in delineating primary tumors and detecting distant metastases because of its low background level in the brain. FAPi PET detected fewer lymph node metastases than FDG PET, but no histologic confirmation exists.
Gu [17]	Detection of primary tumor in patients with HNCUP.	Median: 8.79	Lymph nodes: 9.1 ± 4.7 Bone: 6.96 ± 5.87	n.a.	FAPi PET improves the primary tumor detection rate in HNCUP with negative FDG PET.
Promteangtrong [18]	Comparison of diagnostic accuracy between FAPi PET and FDG PET.	Mean: 19.28 ± 7.45	Lymph nodes: 15.04 ± 10.25 Distant metastases: 13.59 ± 7.64	n.a.	FAPi PET and FDG PET have comparable diagnostic performance for initial staging and recurrence detection in HNSCC patients.
Chen [19]	Assess diagnostic accuracy of FAPi PET.	Mean: 12.74 ± 3.51	Lymph nodes: 4.86 ± 2.51	n.a.	FAPi PET could detect primary HNSCCs and showed superior accuracy in detecting nodal metastases compared to FDG PET.
Wegen [20]	Investigate the use of FAPi PET/CT to detect and delineate HNCs for RT planning.	Median: 5.2	Lymph nodes: 6.8Bone and visceral metastases: 6.5Uncommon sites: 6.0	n.a.	FAPi PET had greater sensitivity and accuracy than FDG PET.
Zheng [21]	Comparison of diagnostic accuracy between FAPi PET and FDG PET.	Mean: 11.3 ± 5.3	Lymph nodes: 7.1 ± 3.6 Bone: 8.3 ± 4.4	Most primary tumors and lymph node metastases showed positive FAP immunostaining; however, FAP immunostaining did not correlate to SUV measured on FAPi PET images.	FAPi PET performed better than FDG PET in detecting intracranial invasion of primary tumors. However, FAPi PET performance was less accurate in lymph node staging.
Jiang [22]	Comparison of diagnostic accuracy between FAPi PET and FDG PET.	Mean: 17.7 ± 7.0	Lymph nodes: 7.83 ± 7.55 Bone: 17.64 ± 10.32	Most primary tumors and lymph node metastases showed positive FAP immunostaining; FAP strongly correlated to SUV measured on FAPi PET images.	FAPi PET performs similarly to FDG PET in detecting primary tumors and has higher specificity in preoperative lymph node staging.

Legend: CT: computed tomography; FAPi: fibroblast activation protein inhibitor; FDG: fluorodeoxyglucose; HNCUP: head and neck cancer of unknown primary; HNSCC: head and neck squamous cell carcinoma; MR: magnetic resonance; n.a.: not available: PET: positron emission tomography; SUV: standard uptake value.

**Table 5 pharmaceuticals-16-01664-t005:** Meta-analysis of primary tumor detection rate.

First Authors [Ref.]	Sample Size	Detection Rate (%)	95% CI	Weight (%)	
				Fixed	Random
Linz [13]	10	100	69.1 to 100	4.76	4.76
Serfling [14]	8	100	63.1 to 100	3.90	3.90
Qin [16]	15	100	78.2 to 100	6.93	6.93
Gu [17]	18	38.89	17.3 to 64.2	7.6	10.97
Promteangtrong * [18]	25	100	86.3 to 100	11.26	11.26
Chen [19]	36	100	90.3 to 100	16.02	16.02
Wegen [20]	15	100	78.2 to 100	6.93	6.93
Zheng [21]	47	100	92.4 to 100	20.78	20.78
Jiang * [22]	67	100	94.6 to 100	29.44	29.44
Total (fixed effects)	241	97.8	95.1 to 99.2	100.00	100.00
Total (random effects)	241	96.3	88.3 to 99.8	100.00	100.00

Legend: CI: confidence interval. *: The DR of primary tumors was assessed only in the staging setting, so the number of patients included in the meta-analysis differs from the overall number of enrolled patients.

## Data Availability

Data sharing is not applicable.

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
