# Peer review of "Diagnostic Accuracy of [68Ga]Ga Labeled Fibroblast-Activation Protein Inhibitors in Detecting Head and Neck Cancer Lesions Using Positron Emission Tomography: A Systematic Review and a Meta-Analysis"

_pharmaceuticals, 2023, doi:10.3390/ph16121664_

Round 1

Reviewer 1 Report

Comments and Suggestions for Authors

Fibroblast-activating protein (FAP) has been a hopeful candidate for molecular imaging of most tumors. Several [ 68Ga]Ga labeled FAP inhibitors (FAPi) have been used to develop tumor radiopharmaceuticals for PET imaging. The objective of this review was to perform a comprehensive evaluation to ascertain the diagnostic performance of [ 68Ga]Ga labeled FAP inhibitors in detecting head and neck cancer (HNC) lesions. In this review, it contained qualitative and quantitative data which emphasize the potential of FAP-targeted PET imaging in identifying primary tumors and distant metastases in patients with HNC. This work is original and is suitable in content for Pharmaceuticals. However, it is in need of some revisions as suggested below:

(1)   On page 3, line 105, “in vivo” should be “in vivo”.

(2)   Since this review focuses on [ 68Ga]Ga labeled FAP inhibitors in detecting head and neck cancer (HNC) lesions, the title should be corrected as “Diagnostic Accuracy of [ 68Ga]Ga labeled Fibroblast-Activation Protein Inhibitors in Detecting Head and Neck Cancer Lesions Using Positron Emission Tomography: a Systematic Review and a Meta-analysis”.

(3)   In particular, there are several format issues in References section. The manuscript needs proof-reading prior to the re-submission.

Author Response

We would like to thank the reviewer for the comments.

Here's our point-by-point answer:

1-2) We rephrased as requested by the reviewer.

3) References were automatically added using a dedicated software (Zotero), so we can't change their format now. Thanks for the advice; we will put effort into their correction during the proof validation step.

Reviewer 2 Report

Comments and Suggestions for Authors

The manuscript Diagnostic Accuracy of Radiolabeled Fibroblast-Activation Protein Inhibitors in Detecting Head and Neck Cancer Lesions using Positron Emission Tomography: a Systematic Review and a Meta-analysis provides a comprehensive evaluation of the imaging methodology utilizing 68Ga-FAPi radiopharmaceuticals for the diagnosis of HNC through a meticulous review of recent literature.

The authors clearly state the motivation behind their study, elucidating the interest in FAP-targeting radiopharmaceuticals due to their enhanced specificity in evaluating the tumor microenvironment across various cancer types. The findings from FAPi imaging demonstrate promising outcomes, particularly in cancer staging and the potential enhancement of radiation therapy planning.

         The choice of studies for review was carefully performed. Based on very fine selection criteria, 11 articles were submitted to qualitative analysis and 9 of those to quantitative analysis. The quantitative meta-analysis is aimed at evaluating the usefulness of 68Ga-FAPi compounds in the detection rate of primary tumors and evaluating lymph nodes in HNC.

         Acknowledging the study's limitations, the authors highlight the constrained sample size and the absence of result validation through 18F-FDG PET and histopathology in a significant portion of the scrutinized studies. Nevertheless, the results derived from the systematic review and meta-analysis underscore the potential of FAP-targeting radiopharmaceuticals in identifying primary tumors and distant metastases in HNC. While acknowledging conflicting literature regarding the utility of Ga-FAPi imaging for lymph node assessment, the authors note the lack of data and advocate for more consistent and effective validation methods.

          The results presented in this study are certainly relevant and offer a thorough evaluation grounded in a literature review of a novel imaging modality. These findings not only provide insights for future investigations into validating the efficacy of FAP-targeting imaging agents but also advocate for prospective clinical trials in this domain.             

Author Response

We are delighted to read the reviewer's comments and thank his/her effort in making a thorough analysis of our manuscript.

In particular, the analysis of lymph node metastases assessment is a pretty complex issue since it might affect our way of analyzing FDG-PET images in clinical practice too; nevertheless, much is yet to be discovered in this field, and we hope to help future research with our article.

Reviewer 3 Report

Comments and Suggestions for Authors

The systematic review and metanalysis presented in “Diagnostic Accuracy of Radiolabeled Fibroblast-Activation 2 Protein Inhibitors in Detecting Head and Neck Cancer Lesions Using Positron Emission Tomography: a Systematic Review and a Meta-analysis” Is a well-designed and organized manuscript. It is well-written and facilitates the reading.

The statistical analysis and explanation of the found results contribute to a better understanding of this emerging field. Further research is needed to validate PET imaging with FAP-targeting radiopharmaceuticals for imaging of primary tumors and metastasis. However, there are some limitations such as the sample size for some papers, the hystological confirmation (in several cases), and the heterogeneity of the studies. These limitations were accurately identified and documented as a result of the performed analysis.

Author Response

We are delighted to read the reviewer's comments.

In particular, as the reviewer advocates, further clinical trials are needed to get more insights concerning this emerging field of research. In this context, one of the preconceived purposes of this systematic review and meta-analysis was to analyze and propose different clinical settings deserving further exploration. 

Reviewer 4 Report

Comments and Suggestions for Authors

In the submitted manuscript, the authors have performed a systematic review and meta-analysis that provides both qualitative and quantitative data highlighting the potential of FAP-targeted PET to identify primary tumors and distant metastases in patients with head and neck cancer. The methodology is well designed and overall the article is well organized and well written. However, the authors conclude that PET imaging with FAP-targeted radiopharmaceuticals needs to be validated to better determine its significance in the assessment of lymph node metastases. As a suggestion, the authors should consider including, as a minor discussion, the heterogeneity of FAP expression in distant metastases, given that one of the functions of FAP is to be expressed to promote migration or distant metastasis, so that once distant metastases are generated, its expression decreases significantly with consequent low or no uptake of the radiopharmaceutical. 

Comments on the Quality of English Language

None

Author Response

We thank the reviewer for his/her comments on our article.

According to the suggestion, we added the following sentence in the discussion section:

As observed in recent literature, FAP expression in the TME is a factor promoting cell migration and subsequent distant metastases onset; once distant metastases are generated, its expression decreases significantly with consequent reduction of the radiopharmaceutical uptake. This observation might explain the heterogeneity in terms of FAP-targeting radiopharmaceuticals uptake in distant metastases, since the PET examination might reveal different lesions in different stages of their evolution. 

Reviewer 5 Report

Comments and Suggestions for Authors

The manuscript "Diagnostic Accuracy of Radiolabeled Fibroblast-Activation 2 Protein Inhibitors in Detecting Head and Neck Cancer Lesions 3 Using Positron Emission Tomography: a Systematic Review 4 and a Meta-analysis" by Rizzo et al perform a thorough meta-analysis of the use of FAPI-PET for HNC, comparing with standard diagnostic tools, including [18F]FDG. The study is well performed and the methodology extensively described and relevant. The conclusion is that no clear answer can be given regarding the diagnostic value of FAPI-PET compared with standard methodology.

Author Response

Thanks to the reviewer for the comments and the analysis of our manuscript.

Since most of the studies included in the systematic review were pilot studies or proof-of-concept papers, the present article could not give definitive conclusions concerning the employment of this novel diagnostic investigation in HNC. However, based on the growing body of literature concerning this emerging field of research, we thought it could be interesting to explore the present literature and advocate for further studies in specific clinical settings.